# GreenNAS: A Green Approach to the Hyperparameters Tuning in Deep Learning

Giorgia Franchini 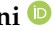

Department of Science Physics, Informatics and Mathematics, University of Modena and Reggio Emilia, 41125 Modena, Italy; giorgia.franchini@unimore.it

**Abstract:** This paper discusses the challenges of the hyperparameter tuning in deep learning models and proposes a green approach to the neural architecture search process that minimizes its environmental impact. The traditional approach of neural architecture search involves sweeping the entire space of possible architectures, which is computationally expensive and time-consuming. Recently, to address this issue, performance predictors have been proposed to estimate the performance of different architectures, thereby reducing the search space and speeding up the exploration process. The proposed approach aims to develop a performance predictor by training only a small percentage of the possible hyperparameter configurations. The suggested predictor can be queried to find the best configurations without training them on the dataset. Numerical examples of image denoising and classification enable us to evaluate the performance of the proposed approach in terms of performance and time complexity.

**Keywords:** neural deep learning; convolutional neural networks; neural architecture search; hyperparameters tuning; performance predictor; GreenAI

**MSC:** 68T07; 68T09

## 1. Introduction

Nowadays, deep learning (DL) has achieved remarkable results in various applications such as image and speech recognition, natural language processing, image deblurring and denoising [1–4], super-resolution [5], autonomous driving vehicles, and image generation [6], among others. However, designing an effective deep learning model requires the tuning of a large number of hyperparameters, such as learning rate, mini-batch size, and features related to the shape and the structure of the network, which is often a tedious and time-consuming process. The crucial distinction between parameters and hyperparameters is that, while the parameters are changed during network training by backpropagation, the hyperparameters must be fixed upstream and remain fixed during the training process. Good or bad configuration of hyperparameters can range from the best result in the literature to complete divergence, even when considering the same methodology with the same data. To address this issue, researchers have explored different methods for automating the hyperparameter tuning, such as neural architecture search (NAS), which has gained significant attention in recent years [7–11].

NAS is the process of automating the design of neural networks by searching for an optimal architecture and other hyperparameters that provide the best performance on a given task. The traditional approach for NAS involves searching of the entire space of possible architectures, which is computationally expensive and time-consuming. To tackle this challenge, researchers proposed the use of performance predictors (PPs) to estimate the performance of different architectures, thereby reducing the search space and speeding up the search process. PPs, once constructed, make the training of new hyperparameter configurations unnecessary, greatly reducing the computational cost from a green artificial intelligence (GreenAI) perspective [12]. PPs are already known in the literature not

only in the NAS context, but also as loss function in blind deblurring problems [13] and memory prediction.

In this paper, we propose a green approach to hyperparameters tuning in deep learning, with the aim to minimize the environmental impact of the NAS process. We argue that the traditional approach to NAS can be resource-intensive and environmentally damaging due to the massive amounts of computation required. Our proposed methodology will leverage the use of PPs to optimize the search process while minimizing its environmental impact. In particular, our approach, an evolution of the work of Franchini et al. [14], is to develop a PP by training only a small percentage of the possible hyperparameter configurations; this PP can be queried to find the best configurations without training them on the dataset.

Many of the methods in the literature in the NAS context are particularly focused on the exploration phase of hyperparameters, taking it for granted that each step will require comprehensive training of the methodology. Performance predictors oppose this view by attempting precisely to make each individual step more efficient. The environmental impact is often not taken into account in these contexts, as modern multinode structures allow multiple trainings to be carried out in parallel, significantly reducing time but with no reduction in energy impact. This is why the proposed method only exploits the power of multinode structures initially, and then continues almost without further calculation, thus imposing considerable energy savings.

The structure of this paper is as follows: in Section 2, some state-of-the-art ideas of the NAS framework based on PPs in a green context will be presented, in Section 3, the proposed novel method, named GreenNAS, with its algorithm will be explained, and in Section 4, an application of GreenNAS to an image denoising problem will allow the evaluation of the effectiveness of the approach. Section 5, in which future extensions of GreenNAS are envisaged, concludes the paper.

## 2. State of the Art

### 2.1. Motivations

In the contemporary landscape of technological advancement, the imperative for sustainable development has become increasingly pressing. Within this context, the emergence of GreenAI, aimed at reducing energy consumption and mitigating environmental impact, stands as a crucial frontier. Developing methods within the realm of GreenAI is not merely a response to environmental concerns, but a strategic imperative for ensuring the longevity and resilience of technological systems. By integrating energy-efficient practices into AI algorithms and frameworks, we not only contribute to environmental preservation but also enhance the economic viability and societal sustainability of AI-driven solutions. Thus, the pursuit of methods tailored to GreenAI represents a pivotal step towards aligning technological progress with environmental stewardship. There are many ways to achieve energy savings, from sharing parameters in multitask learning to compressing language-based models [15]. In any case, whatever the application we are considering, finding a good set of hyperparameters is certainly applicable to any context, and, due to the need to train models over and over again, it is one of the most resource-intensive steps. This is why this paper focuses on NAS methodologies in a green context: GreenNAS.

### 2.2. Other Approaches in the Literature

The state of the art in the context of NAS with PPs includes several methods, such as efficient neural architecture search (ENAS) [16], hierarchical reinforcement learning (HRL) [17], and DARTS [18]. ENAS introduces a controller network that proposes child models, which share parameters with the controller, to perform the search process efficiently. HRL proposes a hierarchical approach where the search is carried out at multiple levels, where the high-level controller selects the sub-architectures to search, and the low-level controller finds the optimal architectures. DARTS proposes a differentiable architecture

search algorithm that optimizes the architectural parameters directly, significantly reducing the search space and improving the related search efficiency.

Exploring in more detail the methodologies that employ NAS in a GreenAI context, we consider different works: in [19], the authors proposed a method for NAS that considers both the accuracy and energy efficiency of the resulting architectures. The paper introduced a search space reduction method that leverages performance predictors to reduce the search space while maintaining high accuracy.

In [20], the authors proposed a scalable and efficient NAS method that uses PPs to accelerate the search process. The authors presented the idea of network morphism, which enables the exploration of a wider range of architectures during the search process by utilizing models that have already been explored.

The paper [12] proposed an NAS method that considers both performance and energy efficiency in the search process. The authors used PPs to reduce the search space and proposed a green-aware metric that evaluates the energy efficiency of the resulting architectures.

In the paper [21], the author introduced a network PP model that estimates the accuracy of different architectures to guide the search process, reducing the explored space.

The paper [22] proposed a graph-based NAS method that uses a PP to guide the search process. The authors introduced a graph-based search space representation that allows for more efficient search and leveraged a PP to improve search efficiency.

Overall, these papers demonstrate the importance of using PPs in the context of NAS to reduce the search space and accelerate the search method while considering the environmental impact of the NAS process.

## 3. Description of the Method

The hyperparameters that are crucial for the performance of an ML methodology can be the learning rate, the mini-batch size, the optimizer algorithm, the layer type, the number of neurons per layer, and the type of layer. The optimal combination of these hyperparameters can greatly impact the performance of the methodology. To find the optimal configuration, a suitable metric can be used to quantify the performance of the network. In this paper, the term "performance" refers to both accuracy and discrepancy. The term accuracy is typically used in a classification context, while discrepancy is used in all those contexts where one seeks to measure a comparison error, such as image reconstruction. The identification of the best hyperparameter configuration is computationally expensive; therefore, an offline method has been developed to predict the performance of a new convolutional neural network (CNN) or, more in general, a new artificial neural network (ANN) corresponding to a hyperparameter configuration. The proposed method employs support vector machines for regression (SVR) [23] and random forest (RF) [24] methodologies to predict the performance of a CNN only from the hyperparameters configuration. It is not necessarily RF or SVR type regressors that have to be used; other types are also used. Obviously, these were chosen because they could be effective regressors and are robust, quick to train, and easily interpretable. The obtained PPs are then used to predict the expected performance of all the possible configurations and find the best ones. This method can be particularly useful for a quality assessment of a new learning methodology.

### 3.1. Phases of the Proposed Methodology

We detail in the following the steps of the proposed algorithm.

1.  The first step is to create a dataset of hyperparameter configurations. This dataset should contain examples where the hyperparameter configuration is labeled with the performance obtained when a numerical stopping condition is reached (maximum 20 epochs in our case). As a measure of performance, one can select the error or the accuracy computed for the test set; hence, it is taken as an average over all examples not seen in the training phase. The dataset should be defined as completely as possible, taking into account both the hyperparameter space and the reasonable execution times in relation to hardware resources. The percentage of samples to include in the dataset

is limited by the available resources, such as time or hardware constraints. Once the samples for the dataset have been chosen, the corresponding CNNs, or another ANN architecture, are trained until stopping condition is reached, and their features and final performances are collected. The training phase is carried out on the training set of a specific problem, while the final performance is evaluated on the testing set. Although the dataset creation phase is expensive, the networks can be trained independently of each other. This allows for high parallelization, making it ideal for multinode infrastructures.

2. In the second phase, the PPs, in our case SVR and RF techniques, are trained on the dataset of the hyperparameter configurations generated at the first step. The input features of the PPs (SVR, RF) are only the hyperparameters of the ANN, while the final ANN performance on the testing set is used as the label. The trained PPs (SVR, RF) can then predict the final performance of an ANN for a specific hyperparameter configuration, even if it is not in the training set. The final predicted performance of an ANN can be obtained also through the HYBRID (HYB) approach, which is the arithmetic mean of the predictions provided by both SVR and RF. This ensemble method increases the effectiveness of the model by reducing the variance. The developed method is computationally convenient since it only requires the definition of a dataset that is well suited to predict the performance of a new methodology solely on the basis of its hyperparametric configuration.

3. In the third stage, the newly trained PPs will be used to compute the predicted performance of the entire hyperparameter space. These scores can be used to select the best hyperparameter configuration or to reduce the size of the exploration space.

In Step 1 of Algorithm 1, we denote by $HS$ the whole space of the hyperparameters. In Step 2, a subsample $\hat{H}S$ is uniformly drawn from $HS$; this subsample is given by those elements that constitute the dataset for the predictors.

---

**Algorithm 1** GreenNAS algorithm.

---

1: Define Hyperparameter Space (HS)
2: Choose at random $n \in \mathbb{N}$ configurations of HS: $\hat{H}S$
3: For each $\hat{H}S_i$, $i = 1, \ldots, n$ calculate the final performance $fp_i\ i = \ldots, n$ with a full training
4: Train the PPs (SVR and RF) with $\hat{H}S$ dataset and $pf$ as label set
5: $S\tilde{V}R = SVR(HS)$
6: $\tilde{R}F = RF(HS)$
7: $H\tilde{Y}B = \frac{SVR+RF}{2}$
8: Sort $S\tilde{V}R, \tilde{R}F, H\tilde{Y}B$
9: Choose the first $m \in \mathbb{N}$ elements of $S\tilde{V}R, \tilde{R}F, H\tilde{Y}B$ for full trainings on a limited HS.

---

### 3.2. Time and Space Complexity of the Proposed Algorithm

A comprehensive analysis of the time and space complexity of Algorithm 1 is crucial to elucidate its environmentally friendly features. This analysis entails a detailed examination of the computational resources consumed at each step of the algorithm and a comparative assessment with conventional methods. In the dataset creation phase (Step 1), although substantial computational resources may be required initially, the algorithm's ability to leverage parallelization on multinode infrastructures optimizes resource utilization, thus enhancing energy efficiency. Furthermore, in the second phase, the training of predictive models such as SVR and RF (Step 2) demonstrates computational efficiency, minimizing redundant computations and optimizing resource allocation, thereby contributing to energy savings. The integration of ensemble methods like the HYBRID approach further enhances model efficiency by reducing variance and ensuring effective resource allocation. Lastly, the use of trained predictive models in Step 3 to compute the predicted performance of the entire hyperparameter space streamlines the search process, conserving computational resources and promoting sustainability in AI research and development. Through a thorough

analysis, Algorithm 1 showcases its potential to contribute to energy-efficient computing practices, aligning with environmentally friendly principles and fostering sustainable technological advancement.

## 4. Numerical Experiment: A Denoising Application

To assess the performance of the suggested technique, we present a numerical experiment in the imaging context. We utilize all three stages of our approach and analyze the achieved results.

We consider the application of a CNN for the purpose of removing noise from images. During the process of image acquisition by various medical or astronomical different tools, Gaussian noise may be introduced into the image. This noise can prevent the ability to accurately read and understand the image. The process of removing artifacts by adding filters after the acquisition of the image can be time-consuming for two reasons. Firstly, filters often require the setup of hyperparameters, which involves several tests. Secondly, they are slow during the inference phase since they are global filters applied to the entire image, making it difficult to parallelize. To address these challenges, we propose the use of a CNN that learns how to remove Gaussian noise from an image. Once trained, the network can quickly clean images from noise, benefiting from the parallelization of convolutional filters. The decision to utilize a CNN for noise removal from images is justified by its proven effectiveness in image processing tasks. CNNs excel in capturing spatial dependencies and hierarchical features within images, making them well suited for noise removal applications. Additionally, their adaptability to diverse data distributions and robust performance across different image types enhance their appeal. While our primary focus is on noise removal, CNNs offer versatility beyond this specific task, with potential applications spanning image classification to object detection. By leveraging the inherent strengths of CNNs, we aim to address the challenge of noise removal while contributing to the broader discourse on the efficacy of deep learning techniques in image processing tasks. In this case we use original, uncorrupted images from the MNIST (Modified National Institute of Standards and Technology database) database [25], which contains handwritten digits commonly used for image classification testing. In our case, we are interested in denoising corrupted images generated from these images, rather than their classification. The dataset used in the experiment consists of grayscale images with pixel values ranging from 0 to 255. After normalization in the range $[0, 1]$, each image is centered in a $28 \times 28$ pixel box. To generate noisy images, the MNIST database is preprocessed by adding random Gaussian noise with a standard deviation of 0.3. Figure 1 shows some examples of original and corrupted images that belong to the train and test sets.

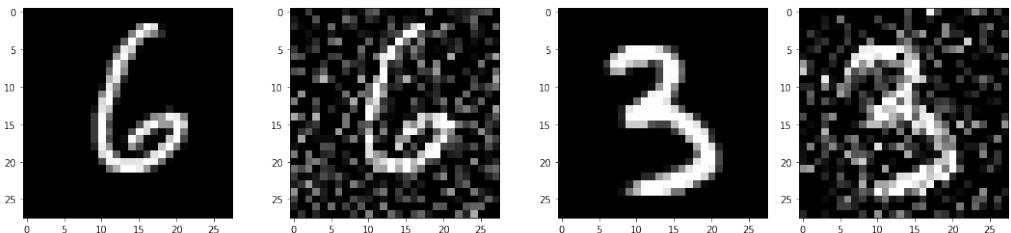

**Figure 1.** Some samples from MNIST database, without and with noise.

To address this image denoising problem, the proposed approach consists of three phases. In the first phase, the CNN hyperparameters are selected and the dataset of examples is generated. The considered hyperparameters include mini-batch size, learning rate, optimizer type, number of filters, and kernel size of a convolutional layer, as suggested in [14]. With regard to the optimizer, the set of optimization algorithms to choose from includes stochastic gradient descent (SGD), SGD with momentum (0.9 momentum hyperparameter), and AdaM. The set of learning rates is $[10^{-3}, 10^{-2}, 5 \times 10^{-2}, 10^{-1}]$ and the one of mini-batch sizes is $[32, 128, 1024]$. The CNN used has 4 convolutional layers, and each

layer can have [1, 4, 8, 16] filters with kernel dimensions of [3, 5, 7]. The hidden layers of the network are convolutions with padding, generating an output with the same size as the input. The activation function used is ReLU.

The entire dataset consists of 746,496 samples, and we choose to estimate $n = 75$ possible configurations. This number was chosen by sampling the space at 0.01% as in [14]. A dataset of 75 examples is sufficient for training standard methodologies, such as RF and SVR [26]. A larger dataset should be used if a neural network is employed as PPs. As consequence, we have to train only 75 CNNs until stopping condition is reached (maximum 20 epochs). The performance measure considered as a label of the built dataset is the mean square error (MSE) computed for the images in the test set. The configurations are chosen according to a uniform distribution in the hyperparameters space. In the second phase, using this dataset of 75 CCNs with the related final performance, we train SVR and RF. At least, in the third phase, with these trained models, we compute all the predicted scores for the entire dataset and we choose $m = 1$ CNN from each method: SVR, RF, and HYB. The choice of $m$ is a trade-off between the desired improvement in performance and the required computational time. The higher $m$ is, the more computational effort we will need for fully training the $m$ CNN for the 3 different performance predictors. The selection $m = 1$ is extreme in the sense of computational savings; it was, in fact, performed to fit better in the GreenAI context.

How to choose hyperparameters in Algorithm 1.

In the various steps of the proposed procedure, and particularly in Algorithm 1, several hyperparameters must be set to achieve the desired effectiveness in the GreenAI context.

In particular, the number of epochs is set according to the considered application. Since the context of interest is that of energy conservation, for a deblurring image problem, a maximum of 20 epochs is well suited [14]. To justify the choice of the number of epochs in the numerical section, we will report the loss function of the best configuration obtained.

Other very important ingredients characterizing the method are the choice of $HS$, i.e., the space of hyperparameters, the selection of the size $n$ of the initial dataset of networks, for the initial dataset, the predictor-related hyperparameters, and the number of $m$ networks among the best predicted for the final trainings. With regard to $HS$, the dependence on the features of the employed architecture is strong. In general, it is always appropriate to consider the hyperparameters that are crucial for the effectiveness of the optimizer, such as the mini-batch size and the learning rate [27]. Other hyperparameters may be those related to the network architecture, such as the size of dense layers or convolutions. In choosing the ranges of these hyperparameters, is necessary to take into account that as these increase, both the training time of individual networks and the prediction time spent on unseen examples will grow. In real-time or near-real-time contexts it makes sense to take these factors into account [28]. In our case, we carefully chose these hyperparameters in a way that places us well in a GreenAI context. In the example considered, the cardinality of the hyperparameter space is a few hundred thousand, but in examples with complex architectures, it can become prohibitively large, even of the order of $10^{16}$ [29].

The choice of hyperparameters forming $HS$ directly determines $n$, in the sense that we choose 0.01% of the entire $HS$. On the other hand, $m$ should always be chosen by weighing the trade-off between accuracy and energy savings. In our application, the extreme case of $m = 1$ was chosen, which implies full training of 3 CNNs, i.e., one for each predictor. This choice should depend on $n$ and on the computational time available.

Finally, the choice of performance predictors and their related hyperparameters is crucial for the effectiveness of the method. In particular, in our case, an RF and an SVR were chosen for several reasons. Firstly, being in a GreenAI context, choosing a deep learning model may not be advisable due to the potential trade-off in the model interpretability it entails; in addition, there are also the need for a large dataset for training the PPs and an increase in the training time for each individual PP. Therefore, the most well-known and robust regressors in the literature were chosen. Of course, these regressors also required careful tuning of the hyperparameters, which was performed automatically

using Optuna [30]. Optuna is a machine-learning-specific automatic hyperparameter optimization software framework. We selected the maximum number of trials (20 in our case) and the hyperparameters to be optimized. The method creates a grid with all the declared combinations and iterates over this grid, evaluating the performance of different configurations. The best configuration is then kept in memory and returned at the end of the process. In our case, for SVR, we decided to optimize the parameter that controls overfitting ($C = \{0.1, 1, 5, 10, 50, 100\}$), the choice of of the kernel type (Gaussian, polynomial, or linear), and the hyperparameters related to the kernels, i.e., the degree in the polynomial $(2, 3, 4)$ or the variance in the Gaussian $(0.01, 0.1, 1)$. In the case of RF (random forest), the following hyperparameters were optimized: the number of trees $(10, 50, 100, 200, 300, 500, 1000)$ and the number of features to consider for each tree $(2, 5, 11)$. The total time required for setting, on an AMD Ryzen 7 processor equipped with NVIDIA GeForce GTX 1650 GPU, the hyperparameters of RF and SVR is less than training a single CNN; therefore, we considered it by adding a unit to the sum of the CNNs that must be trained to perform the method.

In this context, particular attention should be given to steps 5 and 6 of Algorithm 1. Specifically, these steps indicate how the trained SVR and RF models predict the performances of all the networks in *HS*. While this operation may only take a few seconds on an average-powered computer for a dataset of a few hundred thousand examples, it could be a prohibitive task in a space of cardinality of $10^{16}$. In such cases, the hypersurface delineated by SVR, being continuous, differentiable, and convex, can be approached using a gradient-based method to find its unique optimal point. However, in cases where exhaustive execution of steps 5 and 6 of Algorithm 1 is feasible, RF can also be used as a predictor.

### 4.1. Results

During the first phase of generating the CNN dataset, we observe that the selected networks exhibit different levels of performance. In this case, the predictors were trained with the MSE label as a measure of discrepancy, as in this case what is desired is an image as close as possible to the uncorrupted one. In the worst-case scenario, the MSE value, used as the performance measure, exceeds 0.1, resulting in a completely flattened image where both the noise and the image itself disappear. In Figures 2–4, we provide examples of reconstructions, where we compare the original image, the input image with artifacts, the output image, and the difference between the predicted and original images for each configuration.

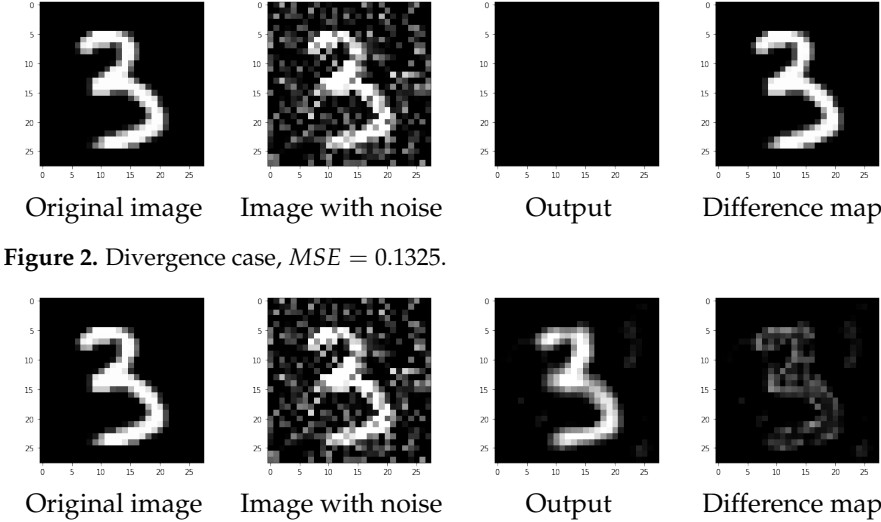

| Original image | Image with noise | Output | Difference map |

**Figure 2.** Divergence case, $MSE = 0.1325$.

| Original image | Image with noise | Output | Difference map |

**Figure 3.** A medium case, $MSE = 0.0601$.

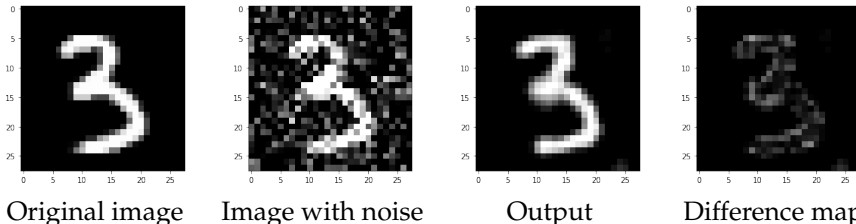

Original image  Image with noise  Output  Difference map

**Figure 4.** A good case, $MSE = 0.0316$.

During the second phase, we use the dataset created in the first phase to train an SVR and an RF. In the third phase, once the predictor has undergone training, it can be utilized to search for the best possible configuration of hyperparameters, which can lead to the production of high-quality reconstructed images. In particular, we use SV, RF, and HYB to predict the MSE on the test set for all the configurations' HS. We sort these performance predictors and we analyze, from a statistical point of view, the best 75 configurations of each PP.

In more detail, the histograms in Figures 5–7 were derived by isolating the best 75 CNNs of each method (SVR, RF, and HYB) based on the predicted performance measured on the test set. The distributions of hyperparameters were studied by considering these best configurations.

In Figure 5, we see how the hyperparameters related to the optimizer are distributed. It is quite evident that almost all of the best predicted CNNs are in the configuration $LR = 0.1$, $OPT = SGD$, $MB = 32$.

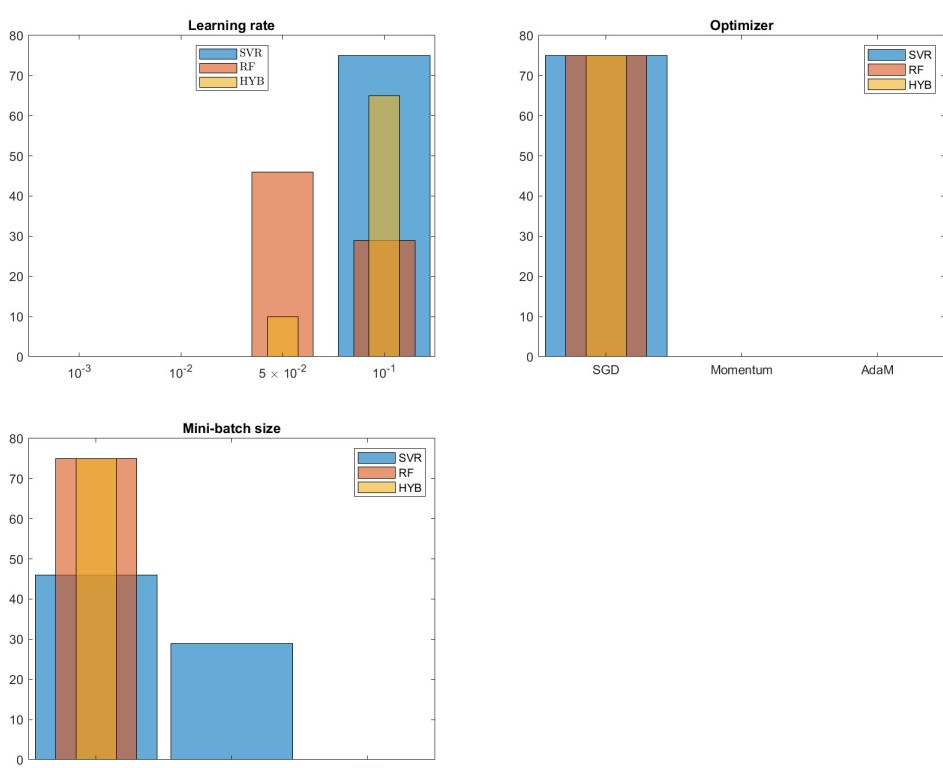

**Figure 5.** Statistics of hyperparameters related to the optimizers for the best 75 predicted CNNs for each of the three methods.

In the same way, in Figure 6, we can observe how the hyperparameters related to the number of filters for each level of the CNN are distributed. In this case, the results are less clear-cut. We can suggest $[4, 8]$ for the first number of kernels, 16 for the second, 4 for the third, and 8 for the last.

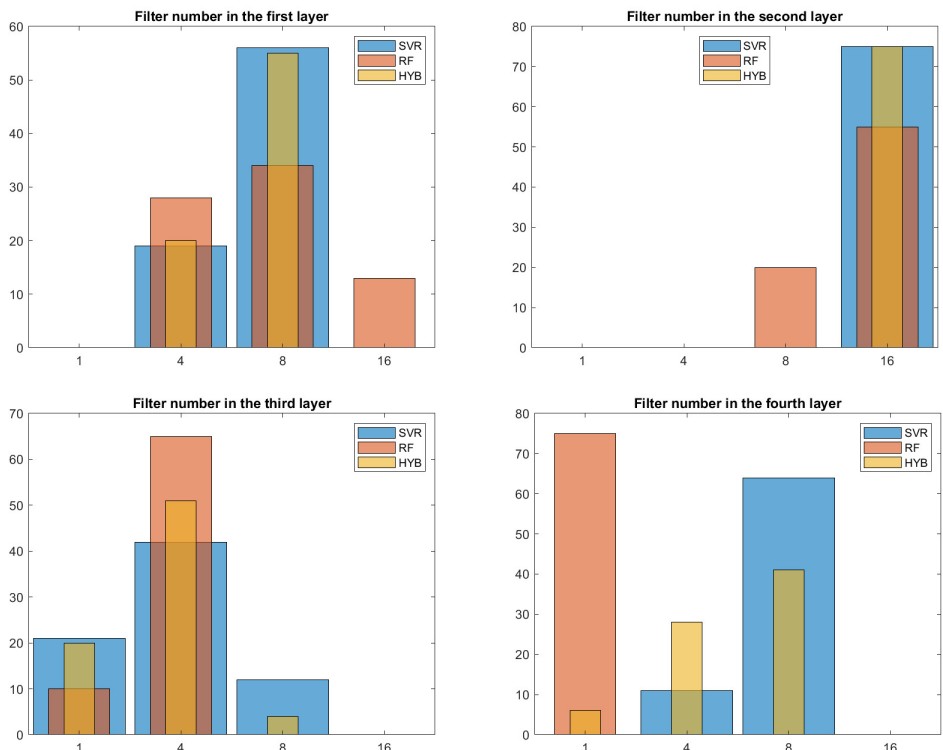

**Figure 6.** Statistics of hyperparameters related to the number of filters in each layer for the best 75 predicted CNNs for each of the three methods.

Finally, in Figure 7, we can observe the distribution of the various kernel sizes; again, a slight preference toward smaller kernel sizes can be observed.

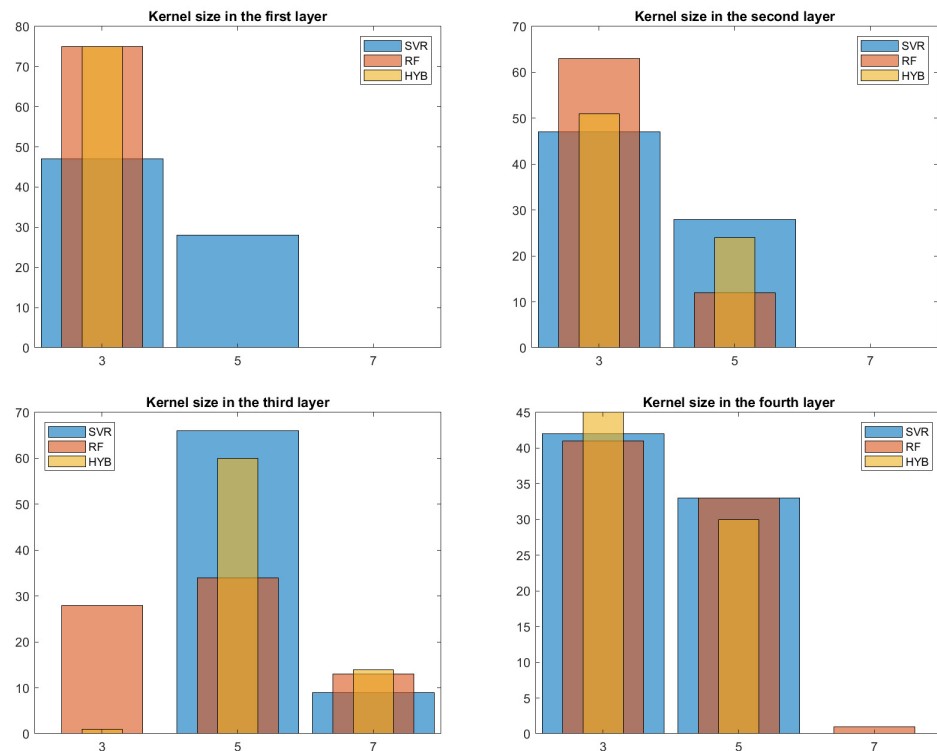

**Figure 7.** Statistics of hyperparameters related to the kernel dimension in each layer for the best 75 predicted CNNs for each of the three methods.

Before proceeding to the use of PPs, it is necessary to analyze their performance. Specifically, after finding the best configuration of hyperparameters with Optuna [30], the predictors were trained on only 70 configurations, isolating 5 of them as test set for the PPs.

Shown in Figure 8 is the performance of the three PPs with respect to these five unseen configurations. As it can be seen, the predictors are not perfect, but they enable the identification of good configurations versus those that will lead to poor performance at the end of the training phase. To make a more rigorous analysis of the PPs, we carried out a specific experiment aimed at looking specifically at the goodness of the PPs. Namely, 75 different CNNs were randomly extracted and divided into train set (80%) and test set (20%). This extraction was carried out in 1000 different draws, thus producing 1000 different train and test sets consisting of 75 different configurations as input and their final performance as output. At this point, the three different PPs were trained for each dataset and we report the average error and the related standard deviation. For the SVR-based predictor, the RMSE (root mean square error) is 0.0355, while the standard deviation is 0.0013. On the other hand, the version with RF achieves an $RMSE = 0.0365$ with standard deviation equal to $1.84 \times 10^{-4}$. The HYB predictor also performs best in the mean version, with an RMSE of 0.0353 and a standard deviation equal to $6.8 \times 10^{-4}$. These results can also be considered partially satisfactory, but must also be evaluated taking into account the small number of configurations used. Having more computational resources available, it is possible to use the method proposed in [14], which also takes into account the performance in the first epochs of the network.

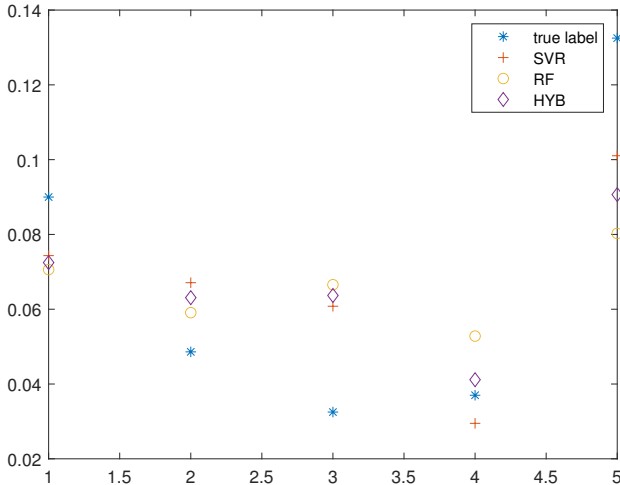

**Figure 8.** Comparison between the PPs performance on 5 unseen configurations.

Based on the results obtained, for any performance predictor, the best CNN can be devised by obtaining the three best CNNs. These three networks were trained until numerical stopping (maximum 20 epochs) on the training set and the performance on the test set was evaluated.

We report in Table 1 the complete configurations of the top three networks obtained by means of the PPs. Furthermore, Figure 9 shows the decreasing behavior of the loss for the test and the train sets in the best configuration; it is evident that the CNN achieves a suitable minimum point. Indeed, the loss function flattens out a lot in the later epochs.

On the other hand, for those configurations with a very bad performance (divergent), the learning process terminates after a few epochs through early stopping as it fails to decrease the loss function. Conversely, there may be some configurations that would require several epochs to obtain a good result, greatly increasing the computational time. The idea is to ignore these particularly slow networks since our context is GreenAI.

**Table 1.** Features and performance for the best 3 predicted CNNs. $NK_i$ and $K_i$ denote the filters number and the kernel size, respectively, of the $i$-th layer, $i = 1, \ldots, 4$.

| OPT | SL | MB | $NK_1$ | $NK_2$ | $NK_3$ | $NK_4$ | $K_1$ | $K_2$ | $K_3$ | $K_4$ | PredMSE | MSE | PP |
|-----|-----|-----|-----|-----|-----|-----|-----|-----|-----|-----|-----|-----|-----|
| 1 | $1 \times 10^{-1}$ | 32 | 8 | 16 | 4 | 8 | 3 | 3 | 5 | 3 | 0.024 | 0.0237 | SVR |
| 1 | 0.05 | 32 | 4 | 16 | 4 | 1 | 3 | 3 | 5 | 3 | 0.0216 | 0.0231 | RF |
| 1 | $1 \times 10^{-1}$ | 32 | 8 | 16 | 4 | 1 | 3 | 3 | 5 | 3 | 0.0203 | **0.0224** | **HYB** |

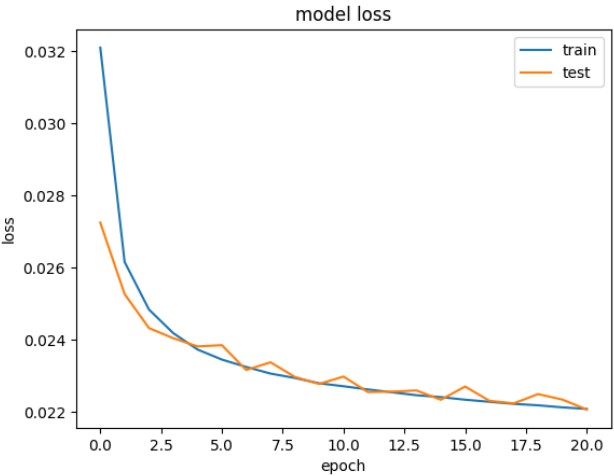

**Figure 9.** Behavior of the loss function for the test and the train sets in the case of the best configuration obtained by the method.

Finally, Figure 10 shows, in addition to the actual performance on the test set of the best three networks found by the PP, the performance of the best ten networks of the original dataset.

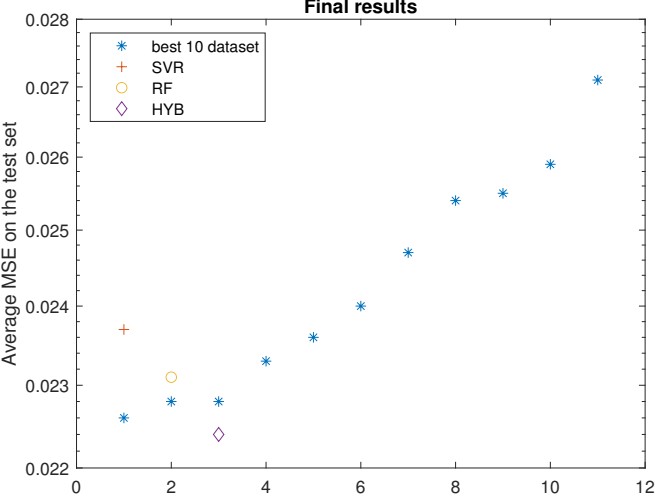

**Figure 10.** Comparison between the best samples of the dataset and the best CNNs found from the PPs.

As can be seen in the last figure, the best network found by the PP HYB achieves the best result of all, including the best in the dataset. The best networks predicted by SVR and RF also perform very well, showing a performance comparable to the best samples in the dataset. We highlight that these results were achieved by training for 20 epochs only on 78 networks, that is, about 0.01% of the size of the hyperparameter space. We also emphasize that the entire process of training, testing, and prediction of PPs took about the same time and resources required for a single CNN training epoch.

In Figure 11, we can see the reconstruction of the best CNN determined by HYB PP on an element of the test set. As we can see, the reconstruction is very clear and only few pixels are visible in the difference map.

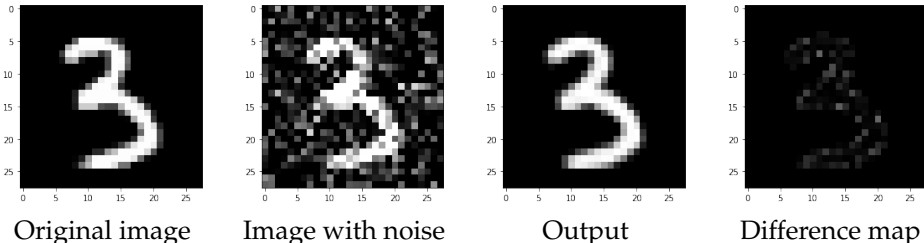

| Original image | Image with noise | Output | Difference map |

**Figure 11.** The best case from HYB PP, $MSE = 0.0224$.

Comparison with other grid search methods without the use of PPs.

In the literature, there exists a variety of NAS methods based on the use of PPs and other techniques that address the problem following different ideas. A comparison of the proposed technique with the state of the art is beyond the scope of this paper, but a first comparison can be performed with grid search type methods. We highlight that grid search methods equipped with cross-validation techniques can be prohibitive; indeed, they would lead to training each individual hyperparameter configuration multiple times on different data splits. Therefore, we can adopt different ways to execute a comparison with a simple grid search. In the first approach, to keep the computational cost fixed, we randomly extracted 79 configurations from *HS*. In the second case, the grid search was conducted on about one-third of the entire *HS*, thus on 200,000 configurations, similar to a brute-force approach.

In Table 2, we report a comparison of the results obtained and the number of configurations needed for the three methods. GreenNAS refers to the best result obtained with the proposed method (HYB). With *light grid search*, we denote the method using the same number of configurations of GreenNAS, but randomly chosen in *HS*. On the other hand, *heavy grid search* employs many more configurations; in particular, it uses 200,000 configurations, and in this case about 85% of these need 20 epochs to obtain satisfactory results. The table can then be read with computational cost in the second row and performance in the third row. Since MSE is a discrepancy index, a lower value is better. We observe that the best result, albeit by a little, is obtained with the *heavy grid search method*, but at a computational cost more than 250 times higher. When GreenNAS is compared with a grid search method that exploits the same computational resources, it is the proposed method that comes out on top. As many as 200,000 different configurations were trained to conduct the comparison with the method *heavy grid search*; these were then used to demonstrate the robustness of the 75 considered configurations, providing the initial dataset.

**Table 2.** Comparison between different methods.

|  | **GreenNAS** | *Light Grid Search* | *Heavy Grid Search* |
|---|---|---|---|
| Configuration used | 79 | 79 | 200,000 |
| MSE on the test set | 0.0224 | 0.0228 | 0.0221 |

Specifically, 100 different trials were conducted, in which 75 configurations were randomly drawn from among 200,000. With these configurations, the best network was then predicted with HYB and led to convergence. The mean and the related standard deviation of the 100 trials are $0.0223 \pm 0.0016$. This result is coherent with that of the presented experiment; the small variance highlights that the method is particularly robust with respect to the choice of 75 configurations.

Interpretability of the method.

An interesting observation about the use of RF as a PP is based on the fact that this is a particularly interpretable method. Indeed, by focusing on a single tree, we can detect which characteristics are to be considered most important for the regression.

It is very interesting to note that the features related to the optimizer are the closest to the root node: they are, therefore, the most important for the decision tree (Figure 12). In fact, in the first node, the split is performed by considering the type of optimizer (1 SGD, 2 momentum, and 3 AdaM); in the second level, the choice of a value for the learning rate is considered, and in the third, the size of the mini-batch is considered.

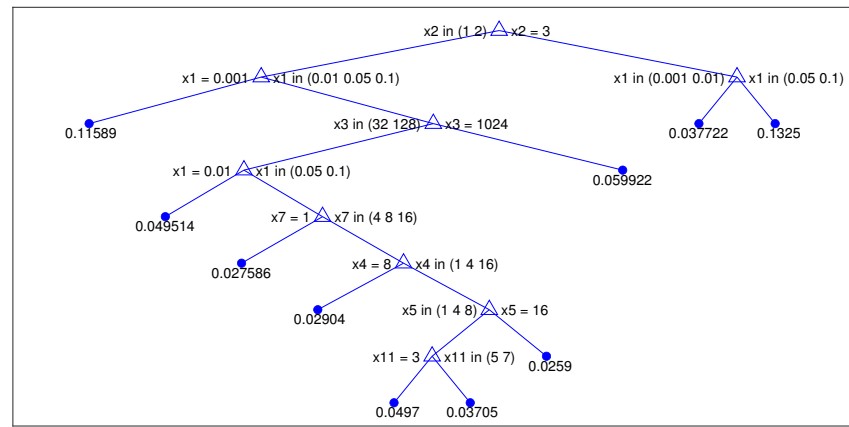

**Figure 12.** One of the binary decision trees in the forest.

### 4.2. A Second Experiment: A Dense Neural Network for Classification

In a second experiment, we consider a multiple classification problem. In particular, we take into account a fully connected network consisting of three dense hidden layers in addition to the input and output layer. The network classifies, through a combination of softmax and categorical cross-entropy, the 10 handwritten digits of the MNIST dataset [25].

In this case, the hyperparameters depending on the optimization procedure will be optimizer type, learning rate, mini-batch size, dimension, and activation function of the three hidden layers. As in the first experiment regarding the optimizer, the set of optimization algorithms to choose from include SGD, SGD with momentum (0.9 momentum hyperparameter), and AdaM. The set of learning rates is $[10^{-3}, 10^{-2}, 5 \times 10^{-2}, 10^{-1}]$, and that of mini-batch sizes is $[64, 256, 512, 1024]$. For the hidden dense layer, we consider $[64, 128, 256, 512]$ for the dimension and ReLU, tanh, and sigmoid as activation functions. The entire dataset consists of 124,416 samples, and we choose to estimate $n = 12$ possible configurations for the initial dataset (0.01%). In this case, the performance used is the accuracy on the test set, which is a set of 10,000 digits never seen during the training phase.

It is well known that the MNIST classification problem by an ANN is a very simple problem; thus, we consider only 15 epochs to evaluate the accuracy of the obtained solution for each different hyperparameter configuration. In the first phase of this experiment, the 12 configurations are chosen with uniform distribution; in the second phase, we build the three PPs using the dataset just created and final accuracies for the labels. In the third phase, we predict all the possible configurations with the three PPs and we choose only the best hyperparameter configuration for each PP ($m = 1$).

### 4.3. Results

To carry out this experiment, we took inspiration from the code proposed in https://colab.research.google.com/github/patbaa/demo_notebooks/blob/master/fully_connected.ipynb#scrollTo=eG11BTseDThc, accessed on 3 February 2020. The technique adopted by this code also provides a baseline for evaluating the obtained results. In particular, the proposed architecture achieves an accuracy of 0.9765 on the test set.

In our case, we therefore trained 12 different networks, chosen at random in *HS*, for 15 epochs each, to create the starting dataset. Using this dataset, the three PPs, RF, SVR,

and HYB, were trained. These PPs were then asked to predict the entire *HS*. Then, the best network was devised and executed to obtain the numerical solution. In summary, the proposed algorithm has the execution cost near equal to 16 networks, which is the executions of 12 networks for the initial dataset, the computational cost equivalent to one network for setting SVR and RF hyperparameters and to train these regressors and, finally, the execution of the 3 final best networks.

We report in Table 3 the predicted and actual results (on the test set) of the original classifier compared with the best configurations returned by PPs. As can be seen, the method returns three configurations better than the one used by the original code.

**Table 3.** Features and performance for the best 3 predicted CNNs. $N_i$ and $A_i$ denote the neuron number and the activation function, respectively, of the *i*-th layer, $i = 1, \ldots, 3$.

| OPT | SL | MB | $N_1$ | $N_2$ | $N_3$ | $A_1$ | $A_2$ | $A_3$ | PredACC | ACC | PP |
|---|---|---|---|---|---|---|---|---|---|---|---|
| AdaM | $1 \times 10^{-3}$ | 512 | 64 | 512 | 128 | ReLU | ReLU | ReLU | 0.9717 | 0.98 | SVR |
| AdaM | $1 \times 10^{-2}$ | 1024 | 128 | 128 | 256 | ReLU | sigmoid | sigmoid | 0.9806 | 0.9803 | RF |
| Momentum | $5 \times 10^{-2}$ | 32 | 128 | 128 | 512 | ReLU | sigmoid | tanh | 0.9799 | **0.9812** | **HYB** |
| SGD | $1 \times 10^{-2}$ | 64 | 512 | 256 | 128 | ReLU | ReLU | ReLU | 0.9637 | 0.9765 | ORIGINAL |

Finally, we report in Figure 13 some of the errors made in the classification phase on the test set of the best network devised by the proposed algorithm.

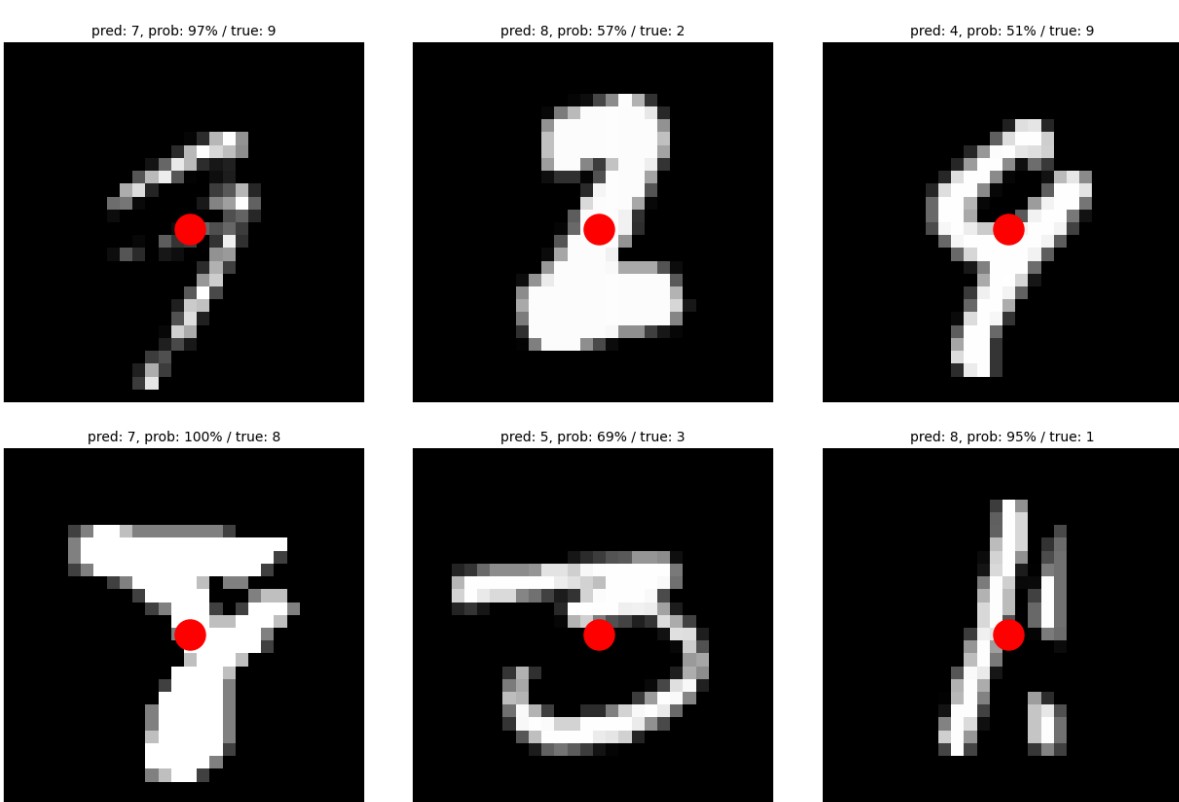

**Figure 13.** Some errors performed by the best configuration.

For each digit, in addition to the image display, we report the predicted label, the probability with which that label is predicted (value returned by softmax), and the true label. We can see in Figure 13 how for several images it is really difficult, even with the human eye, to predict the correct label.

## 5. Conclusions and Future Works

In this paper, we proposed a green approach to the hyperparameter tuning in deep learning, with the aim of minimizing the environmental impact of the NAS process. We argued that the traditional approach to NAS can be resource-intensive and environmentally damaging due to the massive amounts of computation required. Specifically, our approach aims to develop a PP by training only a small percentage of the possible hyperparameter configurations. This PP can be queried to find the best configurations without training them on the dataset.

We presented our proposed approach and provided an algorithm for its implementation. We also discussed the effectiveness of our approach through a numerical application of image denoising.

Our proposed green approach to hyperparameter tuning in deep learning has the ability to significantly reduce the environmental impact of the NAS process, making it a more sustainable and responsible way to design optimal CNN architectures. Future extensions of our approach could include exploring more efficient ways to train performance predictors, investigating the impact of different search spaces on the performance of the PPs, and incorporating other "green" techniques into the NAS process.

Another idea may be to adopt an efficiency threshold for the best PP to adaptively determine $n$ and $m$ according to the desired degree of accuracy.

**Funding:** This work was supported by the "Gruppo Nazionale per il Calcolo Scientifico (GNCS-INdAM), CUP E53C22001930001". The publication was created with the co-financing of the European Union-FSE-REACT-EU, PON Research and Innovation 2014-2020 DM1062/2021.

**Data Availability Statement:** Publicly available datasets were analyzed in this study. This data can be found here: https://git-disl.github.io/GTDLBench/datasets/mnist_datasets/.

**Conflicts of Interest:** The author declares no conflicts of interest.

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
