# Peer review of "GreenNAS: A Green Approach to the Hyperparameters Tuning in Deep Learning"

_mathematics, doi:10.3390/math12060850_

Round 1
Reviewer 1 Report
Comments and Suggestions for Authors
Addressing challenges related to hyperparameter tuning in deep learning models, this paper proposes a sustainable approach to minimize the environmental impact of the Neural Architecture Search process. To evaluate the performance of the suggested approach in terms of accuracy and time complexity, a numerical example focusing on image denoising is presented. Upon comprehensive scrutiny, the paper reveals the following deficiencies:
1. Despite the existence of numerous methods for hyperparameter tuning in deep learning models, the paper lacks clarity in elucidating the motivation behind the proposed model and identifying research gaps.
2. Section 1's structure is overly simplistic, particularly in explaining the concept of the "green approach." The significance of emphasizing environmental impact needs articulation, taking into account the research deficiencies.
3. Sections 1-2 lack an in-depth literature review, making it challenging to discern a robust research motivation.
4. A detailed calculation process is required to elucidate the time and space complexity of Algorithm 1, especially in highlighting its environmentally friendly features.
5. The rationale for utilizing SV, RF and HYB to predict Mean Squared Error on the test set for all configurations HS needs clarification.
6. With fewer than 20 references, the paper's literature review quality is subpar. The article structure should be optimized, potentially by introducing a new section dedicated to literature review.
7. The decision to apply a CNN for noise removal from images necessitates thorough justification. Authors should elaborate on the suitability of this application and its potential applicability in other scenarios.
Comments on the Quality of English LanguageMinor editing of English language required.
Author Response
See the attach.

Reviewer 2 Report
Comments and Suggestions for Authors
The manuscript presents a novel neural architecture search (NAS) method based on SVR and RF trained on a subset of the search space. The methods are well described, the English language is fine, with only few typos that can be easily detected and corrected. I believe the topic to have relevance and interest to the readers, and I have found just some minor issues, as follows:
- the manuscript presents only 18 references, a quantity that seems short for a proper article;
- in the first paragraph of section 2, the term "discrepancy" needs to be defined;
- throughout the manuscript, proper scientific notation shall be used (for example, after figure 1, the learning rates are defined as [1e − 3, 1e − 2, 0.05, 1e − 1], but should be expressed as (10-3, 10-2, 5x10-3);
- in the paragraph after figure 8, the metrics standard deviation and variance are mixed in the same paragraph, and this makes it difficult to compare;
- in line 333, the correct form is "200 000" or "200000". Also, the term "200k" used later should be replaced by "200 000" or "200000";
- in line 377, the correct form is "10 000" or "10000";
Author Response
See the attach.

Round 2
Reviewer 1 Report
Comments and Suggestions for Authors
Accept in present form.